# Survival of the weakest in non-transitive asymmetric interactions among strains of *E. coli*

Michael J. Liao [1,2,5], Arianna Miano[1,2,5], Chloe B. Nguyen[1], Lin Chao[3] & Jeff Hasty [1,2,4 ✉]

Hierarchical organization in ecology, whereby interactions are nested in a manner that leads to a dominant species, naturally result in the exclusion of all but the dominant competitor. Alternatively, non-hierarchical competitive dynamics, such as cyclical interactions, can sustain biodiversity. Here, we designed a simple microbial community with three strains of *E. coli* that cyclically interact through (i) the inhibition of protein production, (ii) the digestion of genomic DNA, and (iii) the disruption of the cell membrane. We find that intrinsic differences in these three major mechanisms of bacterial warfare lead to an unbalanced community that is dominated by the weakest strain. We also use a computational model to describe how the relative toxin strengths, initial fractional occupancies, and spatial patterns affect the maintenance of biodiversity. The engineering of active warfare between microbial species establishes a framework for exploration of the underlying principles that drive complex ecological interactions.

[1] Department of Bioengineering, University of California, San Diego, La Jolla, CA, USA. [2] BioCircuits Institute, University of California, San Diego, La Jolla, CA, USA. [3] Section of Ecology, Behavior and Evolution, Division of Biological Sciences, University of California, San Diego, La Jolla, CA, USA. [4] Molecular Biology Section, Division of Biological Science, University of California, San Diego, La Jolla, CA, USA. [5] These authors contributed equally: Michael J. Liao, Arianna Miano. ✉email: jhasty@ucsd.edu

nter-species interactions form a complex web that drives ecological dynamics[1–5]. Competition, in particular, has been hypothesized as a driving force for the evolution and maintenance of biodiversity within various ecosystems[6–8]. As opposed to a hierarchical competitive structure, previous theoretical studies have shown that species diversity may be promoted by cyclical non-transitive interactions, which describe interactions where there is no single best competitor, but rather the network of species competition resembles a loop[9–15]. The most simplified example of this system can be described as a basic game of rock–paper–scissors. In this system, rock beats scissors, scissors cuts paper, and paper beats rock, resulting in cyclic competition with no hierarchical organization. Ecologies based on this type of interaction have been observed in various natural settings, ranging from desert lizards[12], and costal reef invertebrates[16] to bacterial communities[11,17,18].

Of particular interest within this field of ecology is to better understand the underlying mechanisms that contribute to the stability of these ecosystems[17–20]. While the application of theory in ecology is still limited, the principles of a non-transitive ecology were first demonstrated experimentally using three isolated bacterial strains consisting of a toxin producing, toxin sensitive, and toxin resistant strain[17]. In this natural rock–paper–scissors ecology, the toxin producer could kill the toxin-sensitive strain, the toxin-sensitive strain could outgrow the toxin-resistant strain, likewise the toxin-resistant strain could outgrow the toxin-producing strain. This simple, non-transitive triplet demonstrated that when interactions among the strains remained local, cyclic competition could maintain species diversity. However, because

this study only focused on a short observation duration of 7 days, the ability of non-transitive competition to maintain biodiversity over an ecologically relevant duration remains unclear. Additionally, because this previous study did not provide a characterization of the relative competitive advantages of each strain, it is difficult to relate this system with other non-transitive ecologies. Intuitively, one would expect competitive asymmetry within such an ecology. For example, a toxin-producing strain might outcompete a toxin-sensitive strain at a faster rate than a toxin-sensitive strain could outgrow a toxin-resistant strain. Due to this potential asymmetry of competitive advantages within this system, it is unlikely that coexistence would be maintained over a longer experimental duration.

In this work, we investigate how asymmetric competitive advantages affect a three strain non-transitive ecology, expanding upon the previous experimental work using rationally engineered and well-characterized strains of *E. coli*. Specifically, we engineer three strains of *E. coli* to produce and release three different toxins that kill other members of the same species that lack the production of the corresponding immunity protein. In order to create a one directional rock–paper–scissors competition dynamic, each strain is engineered to produce an additional second immunity protein, providing immunity to one other strain in the ecology. The competitive relationships between each of the strain pairs are characterized in pairwise competition in order to establish relative competitive advantages and identify competitive asymmetry within the system. Using this characterized model microbial system, we explore the outcome of non-transitive competition in a solid growth medium environment (agar) where

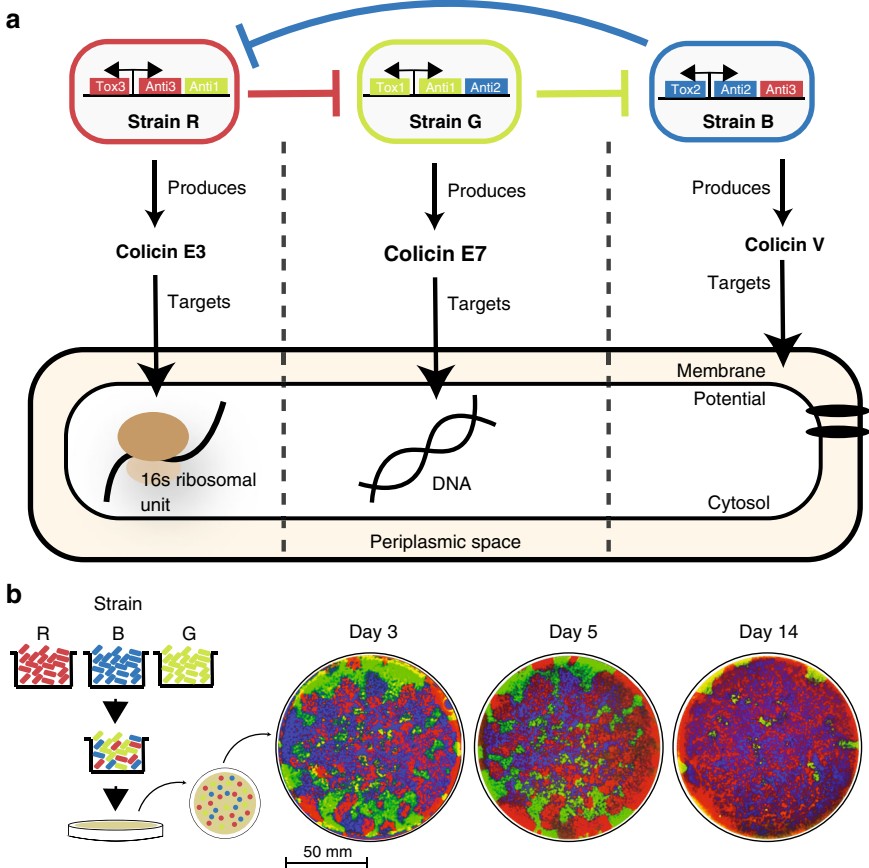

**Fig. 1 Cyclic non-transitive dynamics enable biodiversity. a** Diagram of the engineered ecology strains including one toxin and two immunity genes. Each toxin produced targets a different essential biological component of *E. coli* cells. **b** The three strains were combined at an equal ratio and immediately plated on agar. Each day corresponds to a passage obtained through replica plating. Image stills were obtained by fluorescence imaging. This experiment was executed one time at this density, but it was repeated at multiple densities as shown in Supplementary Fig. 2.

competitive interactions remain local. We find that given equal starting fractions among the three strains, the strain with the weakest competitive advantage consistently dominates the ecosystem. Additionally, we develop a computational model to investigate the relationship between asymmetric competition and parameter space. In particular, we examine the effect of varying the relative competitive strengths among the three strains and find that within a certain parameter space, a steady-state three-strain equilibrium can be reached by the ecology, albeit at different fractional occupancies. Finally, we demonstrate that there are initial conditions such as different patterns of initial distributions that can contribute to, or prevent, the establishment of steady-state coexistence.

## Results

We created a synthetic ecology of three bacterial populations which compete with each other through cyclic non-transitive interactions[21]. In order to create this three-strain ecology where each *E. coli* strain possesses a competitive advantage over another, we used a class of naturally occurring peptide toxins called colicins. These peptides are lethal to certain *E. coli* strains and provide the producing strain with the ability to inhibit growth of competing strains[22]. Among colicins, the mechanisms of action fall within three major categories: disruption of protein production, degradation of DNA, and disruption of the cell membrane[22]. In order to mimic this natural competition diversity, we designed each strain to produce a different colicin that acts through each of the three mechanisms. Additionally, in order to establish a cyclic rock–paper–scissors competitive relationship, each strain was given a second immunity to the toxin produced by the targeted strain (Fig. 1a). In this system, Strain Red (R), contains a plasmid-producing colicin E3 and E3 immunity protein, a fluorescence reporter protein, a lysis protein (for toxin release), and a secondary E7 immunity protein. Strain Green (G) contains the same plasmid structure expressing colicin E7, and the E7 immunity protein, and a secondary Col V immunity protein. Strain Blue (B) contains the wild-type Col V operon consisting of the toxin (*cvaC*), immunity (*cvi*), as well as export proteins (*cvaA* and *cvaB*)[23], and a secondary E3 immunity protein (Supplementary Fig. 1). Due to their different mechanisms of action, inadvertent cross immunity due to structural or functional homology between related colicins was minimized.

The first experiment we carried out to investigate our three strain ecology consisted of mixing the strains, from liquid cultures, at an equal ratio and plating them on a static agar plate environment. The choice to carry out all our experiments on solid media was based on the previously reported observation that local interactions and dispersal promote diversity within a community[17]. After incubation, the bacteria lawn on the agar plate was passaged by replica plating every 24 h to guarantee a constant source of fresh nutrients and uncolonized space to invade (Fig. 1b). We tested a range of initial plating densities spanning a $10^4$ fold change (Supplementary Fig. 2a). We found that the duration of coexistence was inversely proportional to the initial plating densities (Supplementary Fig. 2b). This outcome is attributed to the ability of the lower density cells to establish larger colonies due to the higher availability of free space surrounding them. In this scenario, cells at the colony boundary are able to shield cells in the interior from toxin exposure. Interestingly, we observed one scenario (1–1000 dilution) in which all three strains coexisted for the entire duration of the 30-day experiment (Supplementary Fig. 2c and Supplementary Movie 1). In this scenario, Strain B was nearly able to completely colonize the agar plate, however a small patch of Strain G cells managed to survive the initial attack from Strain R. As a result, Strain G was

able to invade into space colonized by Strain B, and Strain R could invade into the newly acquired area of Strain G. Nevertheless, in the four other scenarios we tested, the system always converged to a single winner, Strain B (Supplementary Movie 2). We wondered whether the survival of Strain G in the 1–1000 dilution was an outlier that occurred because of random distributions in the initial plating, or whether coexistence of this system of three strains was an expected outcome under these conditions. Therefore, we designed an experimental protocol which could guarantee the sequential placement of the three strains in individual colonies which are equally spaced from each other like the intersections of a grid.

In order to achieve precise control of our experimental conditions we used a liquid handling robot to array the three strains into equally spaced grids of varying initial densities. After arraying the cells into grid format on agar plates, the strains were grown for 24 h, then passaged every subsequent 24 h by replica plating (Fig. 2a). The results of this experiment were in agreement with our previous observations. As before, the initial density was inversely related to the duration of coexistence of the three strains, and Strain B was able to take over the agar plate (Fig. 2b and Supplementary Fig. 3). Because competition within microbial communities may be affected by differences in growth rate, we first measured the growth rate of each RPS strain and found that the strains had similar growth rates (Supplementary Fig. 4a). Therefore, we hypothesized that the reason Strain B was consistently winning was due to asymmetry among the potency of the toxins produced by the three competing strains. In order to determine the source of this asymmetry, we characterized the interactions between each pair of competing strains using the same grid pattern arrangement (Fig. 2d). We found that Strain R produced the most lethal toxin, as it was able to completely eliminate Strain G prior to day 5. The next strongest toxin produced by Strain G was able to completely eliminate the Strain B by day 6. Finally, we found Strain B to be the weakest toxin producer, taking 7 days to completely eliminate Strain R (Fig. 2e). The resulting ranking of relative toxin strengths was also demonstrated in kill curve experiments conducted in liquid culture (Supplementary Fig. 4b, c). Therefore, we confirmed the asymmetry in the system, whereby Strain R was shown to be the strongest and Strain B the weakest (Fig. 2f). Following this observation, our hypothesis was that since Strain R was the strongest toxin-producing strain, it was quickly, and completely, eliminating Strain G. As a result, when Strain B and Strain R were the only two left competing, Strain B was eventually able to win due to its immunity and toxicity to Strain R. In this first hypothesis, Strain B would win due to its position as the enemy of the strongest strain (Supplementary Fig. 5a). However, we also considered the counterintuitive possibility that Strain B was consistently winning because it was producing the weakest toxin (Supplementary Fig. 5b). In order to test this theory, we created a new three-strain ecology (RPS-2) in which we reversed the order of the rock–paper–scissors competitive relationship by swapping the secondary immunity genes for each strain. With these new constructs, Strain R2 now attacked Strain B2, Strain B2 attacked Strain G2, and Strain G2 attacked Strain R2. Although we reversed the order of competition by reversing the secondary immunity, the relative toxin strengths remained the same (Fig. 3a and Supplementary Fig. 6). Therefore, while Strain B2 remained the weakest strain, the enemy of the strongest then became Strain G2. If our initial hypothesis of the winner being the enemy of the strongest were correct, we would expect Strain G2 to eventually take over the plate. Alternatively, if "predominance of the weakest" were the driving force, then we would expect Strain B2 to win in RPS-2 as well (Supplementary Fig. 5b). Simultaneously, we developed a lattice-based computational model to simulate

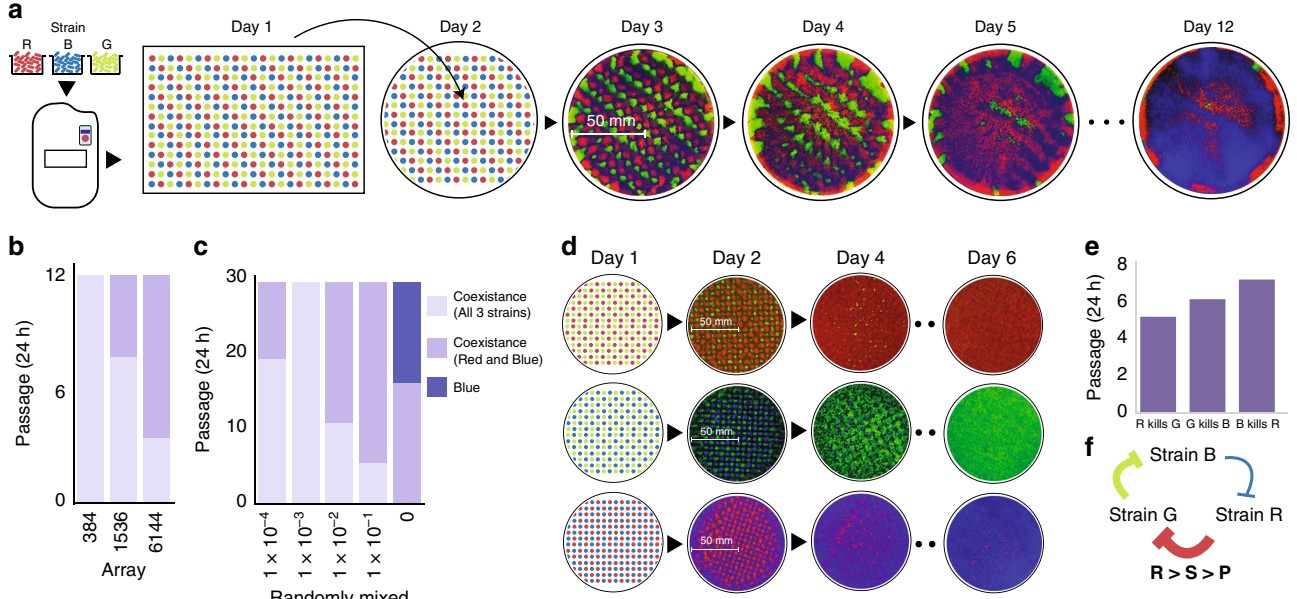

**Fig. 2 Characterization of the ecology and causes of asymmetry. a** Schematic diagram of the method used to create an ordered grid of RGB colonies on agar plate through the use of a liquid handling robot. On the right, images corresponds to passages of the lowest density grid (384). Passaging of plates was done by replica plating. This experiment was executed one time at this density and it was repeated at a higher density as shown in Supplementary Fig. 3. **b** Quantification of strain coexistence as a function of time for three different initial densities plated in grid format. **c** Quantification of strain coexistence as a function of time for five different initial densities plated in a random distribution. **d** Paired competition on agar plates starting from an initial grid at 1536 density. **e** Quantification of time to takeover for each competing pair. **f** Diagram illustrating asymmetry in toxin strength in the RPS ecology. Strain R produces strong inhibition of Strain G, Strain G produces intermediate inhibition of Strain B, and Strain B produces weak inhibition of Strain R.

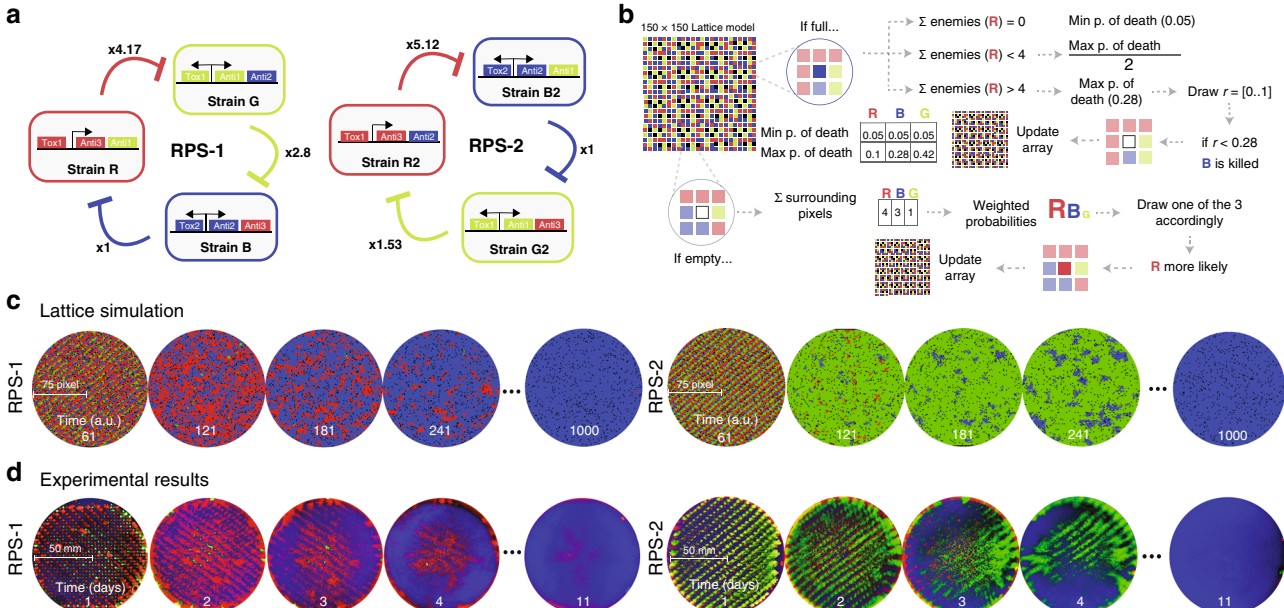

**Fig. 3 Characterization of reversed RPS ecology and computational model. a** Diagrams of RPS (RPS-1) and RPS reversed (RPS-2) in comparison. The numbers represent the relative toxin strength obtained from kill curve experiments. Source data are provided as a Source Data file. **b** Pseudocode illustrates the rules behind the lattice model simulations. **c** Comparison of model simulations of RPS-1 (left) to model simulations of RPS-2 (right) with a grid of density corresponding to 1536. **d** Comparison of experimental results of RPS-1 (left) to experimental results of RPS-2 (right) with density 1536. All lattice simulations plots were simulated in a square lattice grid and subsequently trimmed into circular shapes to ease the comparison with experimental results. This experiment was executed one time at this density and it was repeated at a lower density as shown in Supplementary Figs. 3 and 8.

competition of the three strains in a $150 \times 150$ square grid (Fig. 3b). With the parameter values established in our kill curve experiments, simulations of our computational model agreed with our previous experimental results demonstrating that Strain B should outcompete the other strains (Supplementary Movie 2).

The simulations also confirmed our observation regarding the relationship between duration of coexistence and initial density (Supplementary Fig. 7). Furthermore, when simulating the RPS-2 system, the model predicted that the eventual winner of the RPS ecology would still be Strain B, confirming the idea of

"predominance of the weakest" (Fig. 3c and Supplementary Movie 4), a theory that had been hypothesized by previous theoretical works[24–29]. As predicted, we found experimentally that the final winner of the RPS-2 ecology was indeed Strain B2 (Fig. 3d, Supplementary Fig. 8 and Supplementary Movie 5). This outcome occurs because the weakest strain allows its target (Strain R in RPS-1 and Strain G2 in RPS-2) to rapidly expand and eliminate the third strain in the system (Strain G in RPS-1 and Strain R in RPS-2). When only two strains are left, the "weakest" strain is then able to slowly take over due to the rock–paper–scissors dynamics (Supplementary Fig. 9). While we were enthusiastic to experimentally demonstrate the theory of "predominance of the weakest", we still wondered about the scenario we observed in which all three strains were not fully eliminated, and seemed to have developed stable existence. We therefore hypothesized that there must be alternate possible outcomes that arise based on probability for a given set of toxin parameters.

To further explore this concept, we investigated the relationship between toxin strength and the system steady state. We used our computational model to do a parameter sweep of toxin production strengths. For each combination of parameters, we ran 100 simulations. We found that "predominance of the weakest" was not always the most probable outcome. When keeping the toxin strength of both the weakest and strongest toxins fixed while varying the toxin strength of the intermediate strain, the most probable steady-state outcome was dependent on the intermediate toxin strength (Fig. 4a). The model shows that if the toxin strength of the intermediate strain (Strain G) is close in value to the toxin production strength of the weakest toxin

producer (Strain B), the most probable outcome is that the intermediate wins (strain G) (Fig. 4b). On the other hand, when the toxin production strength of the intermediate strain falls within an intermediate range, the model predicts the possibility of stable coexistence between the three strains at different fractional occupancies (Fig. 4c and Supplementary Movie 6). Finally, if the toxin production strength of the intermediate strain is close in value to the strength of the strongest toxin producer (Strain R), then the weakest strain (Strain B) always wins as predicted by the hypothesis of "predominance of the weakest" (Fig. 4d). Interestingly, we did not observe any case in which the strongest strain (Strain R) won. When compared to our previous experimental results, our experimental parameter values fall within the parameter space that predicts two possible outcomes. The most probable scenario is that Strain B wins, but there is also a small probability of the establishment of coexistence. In agreement with the model, we observed that on the plate that established coexistence, the three strains maintained different fractional occupancies, with Strain B having the highest fractional occupancy. These results highlight the value of the computational model for examining a wide variety of scenarios that would be too difficult or time consuming to cover experimentally. To this end, we wanted to investigate how different initial fractional occupancies would effect the steady-state outcome. We found that, in grid format, the final outcome is highly influenced by the abundance of the weak strain, where an inverse relationship exists between the starting fractional occupancy of the weakest strain and its probability to takeover (Supplementary Fig. 10). We also explored how different patterns of initial strain distribution would affect the steady-state outcome of the competition in relation to toxin

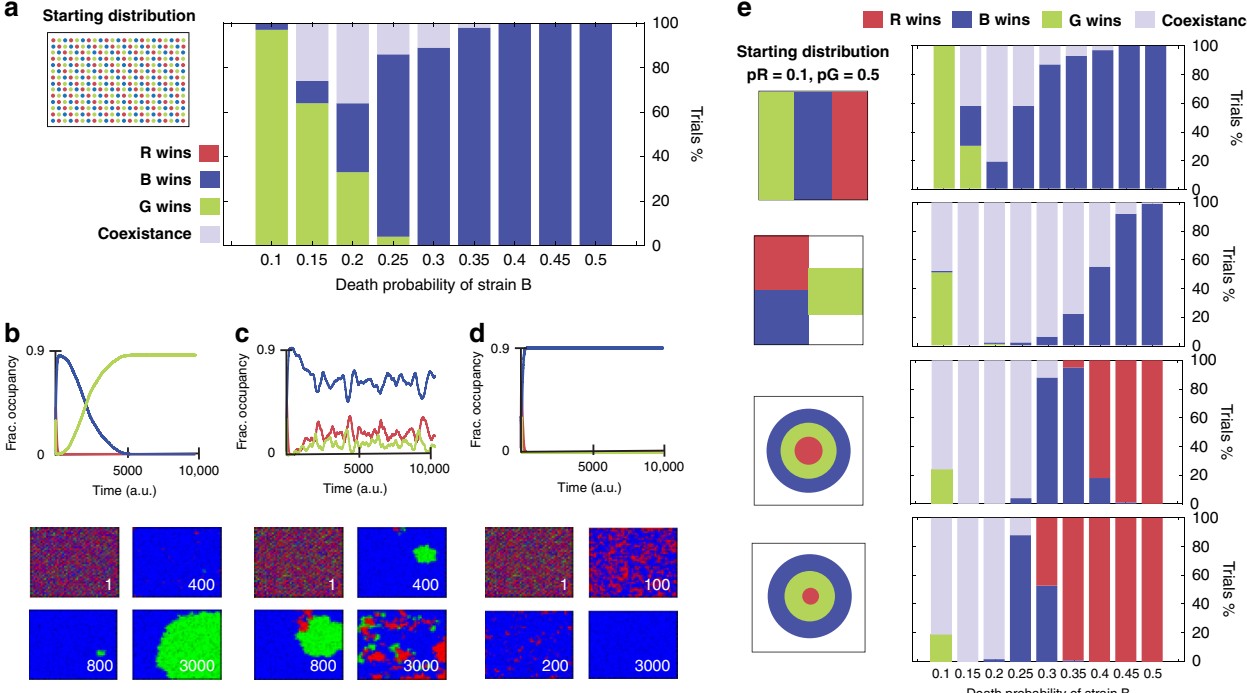

**Fig. 4 Model simulations exploring spatial patterns and parameter space. a** In this simulation the starting condition is an ordered array of alternating strains. The bar plots shows the outcome of 100 trials for multiple parameters of Pb (Probability of death of strain B). For all simulations, the probability of death of strain R (0.1) and strain G (0.5) are kept constant. **b** Time series representing the possible scenario where strain G wins (left-hand side of bar plot in part **a**). Below, the corresponding image stills at multiple time points. **c** Time series representing the possible scenario where strain R B G coexist (left-hand side of bar plot in part **a**). Below, the corresponding image stills at multiple time points. **d** Time series representing the possible scenario where strain B wins (right-hand side of bar plot in part **a**). Below, the corresponding image stills at multiple time points. **e** We generated the same bar plots described in part **a** with different geometries as initial conditions. The bar plots shows the outcome of 100 trials for multiple parameters of Pb (Probability of death of strain B). For all simulations, the probability of death of strain R (0.1) and strain G (0.5) are kept constant.

strength. Using the computational model, we simulated multiple scenarios with common spatial patterns, such as stripes, isolated clusters, and concentric circles (Fig. 4e). We found that stripes led to similar results compared to the previously discussed case with the grid pattern in which the weakest strain is able to mostly dominate the system (Supplementary Movie 2). On the contrary, having the strains separated in three different clusters resulted in a significant increase in the parameter ranges that could lead to coexistence (Supplementary Movie 2). As before, in both scenarios we did not observe any case in which the strongest strain (Strain R) could win. On the other hand, when initially distributed in the pattern of concentric rings, it was possible for the strongest strain (Strain R) to take over the plate (Supplementary Movie 2). This only occurs when the strongest strain is placed in the inner circle, shielded from the weakest strain (Strain B). This way, if the intermediate strain in the middle manages to kill off the outer strain before Strain R reaches it, then the strongest strain can win. As expected, we also demonstrate that this outcome is dependant upon the ratio between the ring dimensions and the rate of takeover of each strain (Supplementary Movie 2).

## Discussion

Bacterial communities occupy a myriad of diverse niche environments, playing important roles in processes ranging from nutrient recycling[30,31] to the regulation of human health[32–34]. While the ability to engineer robust bacterial communities could lead to major advancements in fields, such as recycling, sustainability, and healthcare, the mechanisms underlying species diversity and stability are still not yet well understood. Although cooperative interactions between the species comprising such communities contribute to ecological stability[35–37], it is generally accepted that competition is the guiding force[19,32,38–40]. However, due to the complexity of natural ecologies, these types of competitive interactions are difficult to isolate or quantify in nature. Our study demonstrates the feasibility of using an engineered synthetic ecology to simplify complex community relationships in order to study underlying mechanisms that may lead to community stability and the maintenance of diversity. Due to the immense species diversity and wide range of competitive strategies organisms employ in nature[19,40,41], we hypothesized that natural competitive dynamics are likely to be unbalanced[42,43]. Unlike a perfectly balanced game of rock–paper–scissors in which each of the three species could kill each other at an equal rate, we focus on characterizing an asymmetric system in which the relative competitive advantages among each predator–prey pair varies. While previous studies focused on the coexistence of non-transitive communities over a relatively short timeframe[17], they do not allow the system to reach steady state, in which case it is likely that only one species would remain. Here, we demonstrate that intransitivity fails to promote biodiversity over a long time horizon when the relative competitive advantages are imbalanced[44]. Therefore, we believe that an asymmetric non-transitive ecology is a useful base model to study complex interactions among competing bacterial species. Using our three-strain ecology we experimentally demonstrate that a uniformly distributed asymmetric game of rock–paper–scissor is most likely to be won by the weakest species[24–29]. Interestingly, we show that an asymmetric ecology can develop steady-state coexistence, and that the relative toxin strengths among the three species dictate the extent of the coexistence space. Counterintuitively, under the same initial conditions of uniform distribution, our models predict that the producer of the most lethal toxin never wins. As opposed to pairwise competition, where the producer of the strongest toxin has a competitive advantage[6], the producer of the strongest toxin

in non-transitive communities are at an evolutionary disadvantage. This could be an important selective force against continuous evolution of ever-more lethal warfare chemicals in microorganisms, resulting in increased diversity of chemicals that are constrained to a specific toxin strength parameter space. The role that toxin-mediated competition plays in community stability may also explain the observed relative abundance of membrane targeting DNA, or ribosome targeting bacterial toxins among bacterial communities[45]. Finally, we also observe that the steady-state outcomes of the system can be altered by changing the initial strain distribution patterns. For example, we find that separate blocks in a triangular conformation can greatly expand the parameter space for coexistence, supporting the idea that spatially separated niches are more likely to sustain biodiversity[46–48]. On the other hand, strains initially distributed in concentric circles can enable the strongest toxin producer to win. These results demonstrate that many factors need to be considered if the goal is to design stable synthetic ecologies in an environment where interactions are local[6]. Overall, this study provides a mathematical model and engineering framework to study competitive interactions, gain mechanistic insight, and ultimately, predictive power that can be used as a guide to design stable communities.

## Methods

**Strains and plasmids.** Our strains were cultured in lysogeny broth (LB) media with 50 µg ml$^{-1}$ spectinomycin for the WT and TP strains, in a 37 °C shaking incubator. The plasmids used in this study are described in (Supplementary Fig. 1). The colicin E3, Im3, colicin E7, Im E7, and E1 lysis genes were taken from previously used plasmids[21] using PCR and assembled with Gibson assembly[49]. The 4.5 kb colicin V expression cassette comprising *CvaA*, *CvaB*, *CvaC*, and *cvi* was isolated from the pHK11 plasmid from the wild type colicin V strain ZK503 by PCR[23]. All plasmids were transformation into DH5α (Thermofisher) chemically competent *E. coli* and verified by Sanger sequencing before transformation into *E. coli* strain MG1655. The strains used in this study are described in (Supplementary Table 1). The gene sequences used in this study are described in (Supplementary Table 2).

**Growth rate.** For growth rate experiments, the appropriate *E. coli* strains were seeded from a −80 °C glycerol stock into 2 ml LB and the appropriate antibiotics and incubated in a 37 °C shaking incubator. After cells reached an OD600 of 0.1, 1 ml culture was added to a 125 ml Erlenmeyer flask containing 25 ml fresh media with appropriate antibiotics and left shaking at 270 rpm. Once the samples reached an OD600 of 0.1 samples were taken every 20 min and measured at OD600 using a DU 740 Life Science Uv/vis spectrophotometer.

**Toxin validation.** To prepare colicin lysate, the colicin E3 *E. coli* strain was seeded from a −80 °C glycerol stock into 2 ml LB and incubated in a 37 °C shaking incubator. After cells reached an OD600 between 0.4 and 0.6, 1 ml of the grown culture was collected in a 2 ml Eppendorf tube and two cycles of incubation at 98 °C for 5 min followed by 10 min at −80 °C were performed. The resulting media was then filtered and collected using a 0.22 µm syringe filter. For toxin co-culture experiments, wild type MG1655 *E. coli* strains were seeded from a −80 °C glycerol stock into 2 ml LB and incubated in a 37 °C shaking incubator. After cells reached an OD600 between 0.2 and 0.4, 5 µl culture was added to 200 µl fresh media in a standard Falcon tissue culture 96-well flat bottom plate. Additionally, 5 µl of the purified colicin lysate was added to each well. Cultures were grown at 37 °C shaking for 19 h and their optical density at 600 nm absorbance was measured every 5 min with a Tecan Infinite M200 Pro.

**Plate passage experiments.** Each *E. coli* strain was seeded from a −80 °C glycerol stock into 5 ml LB with 50 µg ml$^{-1}$ spectinomycin. After growth for 8–12 h at 37 °C in a shaking incubator, the culture was diluted 100-fold into 25 ml of the same medium in a 50 ml Erlenmeyer flask and grown until reaching an OD of 0.4 (Plastibrand 1.5 ml cuvettes were used). Strains were then diluted 1:10, 1:100, 1:1000, and 1:10,000. 20 µl of each strain was then plated into separate regions of 100 × 15 mm Petri dishes containing LB agar and 50 µg ml$^{-1}$ spectinomycin. The strains were then mixed using 10 glass beads and incubated overnight at 37 °C for 24 h. Plates were passaged every 24 h by replica plating onto a fresh agar plate containing the appropriate antibiotics.

**Liquid handling robot.** As before, each *E. coli* strain was seeded from a −80 °C glycerol stock into 5 ml LB with 50 µg ml$^{-1}$ spectinomycin. After growth for 8–12 h

at 37 °C in a shaking incubator, the culture was diluted 100-fold into 25 ml of the same medium in a 50 ml Erlenmeyer flask and grown until reaching an OD of 0.75. Upon reaching this OD, 5% glycerol was added to each culture and 45 μl of each prepared culture was then transferred into a single 384-well labcyte plate. To plate the grid array we used a Labcyte Echo liquid handling robot to transfer 2.5 nl volume from the source well on the labcyte well plate onto SBS-format PlusPlates containing 42 ml LB agar and 50 μg ml$^{-1}$ spectinomycin. The transferred cells were arrayed according to 384, 1536, and 6144 well plate formats in order to create varying densities. The cells were then incubated overnight at 37 °C for 24 h. Colonies were transferred by replica plating from the SBS-format PlusPlates onto a standard 100 × 15 mm Petri dish containing LB agar and 50 μg ml$^{-1}$ spectinomycin. Plates were passaged every 24 h by replica plating onto a fresh agar plate containing the appropriate antibiotics.

**Kill curve**. To prepare colicin lysate for each strain, the appropriate strains were seeded from a −80 °C glycerol stock into 5 ml LB and incubated in a 37 °C shaking incubator overnight. The overnight cultures were then passaged 1:100 into 2 ml LB and incubated in a 37 °C shaking incubator until reaching an OD600 of 1.0. 1 ml of the culture was then collected in a 2 ml Eppendorf tube, centrifuged at 21,130 rcf for 10 min and the supernatant was passed through a 0.22 μm syringe filter. For the kill curve assay, the appropriate strains were grown in a 37 °C shaking incubator to an OD600 between 0.3 and 0.4 in 5 ml LB in a 25 ml flask. 500 μl of the corresponding colicin lysate was then added to the flask. CFU measurements were taken by plating serial dilutions ($n = 3$) and counting colonies. The first time point was taken immediately before each colicin was added, then every 10 min afterwards. For analysis, we took the difference of five time points (40 min) that corresponded to the maximum change in total alive cell count divided by the initial CFU count of the initial time point in order to calculate the percent cell death. The relative toxin strengths were then found by taking the ratios between the percent cell deaths for each strain pair.

**Image processing**. For plate imaging a Syngene PXi fluorescent imager was used. Strains producing sfGFP were captured using the Blue LED Module for excitation and SW032 emission filter, and strains producing mKate2 were captured using the Red LED Module excitation source and Filt 705 emission filter. Images were processed using Image J. Images were converted to 8-bit and background subtracted. In order to assign the false color blue, we inverted the fluorescence values of the sfGFP image using the math function $v = -v + v(\text{mean})$. We then took the difference from the mKate2 image in order to create a mask of the negative space from both GFP and RFP. This image was assigned as the "Blue" channel for composite images.

**Modeling**. We developed a lattice-based model in Matlab to simulate the competition dynamics between the three strains RGB. The model was based on similar principles previously described in the literature[17]. The lattice is a 150 × 150 regular square lattice with zero boundary conditions. This means that the edges of the lattice are set to zero (absorbing boundary conditions), simulating the physical boundary of the Petri dish that prevents cells from expanding beyond it, as well as the effect of disregarding cells beyond the boundary of the replica plating. Therefore, the four edges of the square are kept at a value of zero (no cells can grow/expand in that direction) and the grid is updated only for the internal pixel (pixel 2 to $N - 1$ on all sides). The simulations in Figs. 2 and 3 were obtained by starting the lattice with a grid array of alternating strains. The remaining lattice points are classified as "empty space". The probability of death for each competing strain is associated to the relative potency of its enemy's toxin. pR, pG, and pB refer to the maximum probability of death of strain R, G, and B, respectively. For each time loop, the lattice array is scanned pixel by pixel (ignoring boundary pixels which have a fixed value of 0) and is updated according to two main rules as shown in Fig. 3b. If the pixel considered is empty, the algorithm takes into account the relative occupancy of the eight neighboring pixels for the three strains R, G, or B. Three probabilities are calculates as the sum of locations occupied by each strain divided by the total neighboring locations (8). Finally, the Matlab function *randsample* is used to choose how to fill the spot according to the previously calculated probabilities. This process simulates expansion due to growth. On the other hand, if a given location is full, the strain is killed with a thresholded probability that is dependent on the number of surrounding enemies present. If the number of enemies is below 4 (corresponding to being ≤half surrounded), the probability of death is capped at half the maximum probability of death for the given strain. On the other hand, if the number of surrounding enemies is >4, the probability of death corresponds to the maximum probability associated to the given strain. In addition, we set a baseline probability of death = 0.05 which cumulatively represents the random death of cells and their removal due to the replica plate passaging. For the simulations shown in Fig. 4, each simulation was run for $t = 10,000$ and each parameter set was simulated 100 times. Each time point corresponds to a reproduction event (around 25 min). Therefore, the time chosen to investigate steady-state dynamics corresponds to about 170 days, which we established to be a long enough interval both computationally and biologically. In terms of spatial parameters, the distance between two consecutive pixels represents roughly 1 mm on the agar plate. Therefore, the entire grid represents a square with a side of about

10 cm. The relationship between grid densities on the agar plates compared to the model are illustrated in Supplementary Fig. 7.

**Reporting summary**. Further information on research design is available in the Nature Research Reporting Summary linked to this article.

## Data availability
Authors can confirm that all relevant data are included in the paper and/or its supplementary information files. In addition, Source Data are provided with this paper. Plasmids and bacterial strains are available from the corresponding author upon request. Source data are provided with this paper.

## Code availability
All code is available on github at the following link https://github.com/armiano/Survival-of-the-weakest-in-non-transitive-asymmetric-interactions-among-strains-of-E.-coli.

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

## Acknowledgements
We thank K. Chen and Professor S. Fraley for help imaging and providing the Syngene PXi fluorescent imager used to image agar plates. This work was supported by the National Institute of General Medical Sciences of the National Institutes of Health (grant no. R01GM069811). M.J.L. is supported by the National Science Foundation Graduate Research Fellowship (grant no. DGE-1650112).

## Author contributions
M.J.L., A.M., L.C., and J.H. contributed to the development of the project. M.J.L., A.M., and C.N. constructed the plasmids and strains and analyzed results. M.J.L., A.M., and C.B.N. conducted the experiments. A.M. conducted the computational modeling. M.J.L. and A.M. prepared the figures, and M.J.L., A.M., L.C. and J.H. prepared the manuscript.

## Competing interests
J.H. has a financial interest in GenCirq. The remaining authors declare no competing interests.
