## [Peer Review File · Nature Communications]

Reviewers' Comments:

Reviewer #1:

Remarks to the Author:

Liao et al. used engineered E. coli strains engaged in different types of colicin-mediated warfare (inhibiting protein production, digesting genomic DNA, disrupting membrane) to study cyclic Rock-Paper-Scissors interaction networks in a spatially-structured environment. As expected, the coexistence of the three strains depended on initial densities (lower density means more spatial structure and longer period of coexistence). Surprisingly, the weakest strain often dominated. This was because the weakest strain allowed its target to expand and eliminate the weakest strain's enemy, and afterwards, the weakest strain could destroy its target and dominate the entire community. The outcome of species coexistence also depended on spatial configurations. This work complements our current understanding of cyclic interactions in a spatially-structured environment. The work is interesting, and writing is clear. I have no major concerns.

Minor:

1. Fig 4d legend: should it be strain "B" wins?
2. In Discussions, "Surprisingly, we observe that xxx": this statement seems to be opposite to Fig 10?

Review prepared by Wenying Shou

Reviewer #2:

Remarks to the Author:

The present study shows that, in a non-transitive competitive interaction involving chemical warfare, the community is more likely to be dominated by the species (strain) with the lowest toxic strength. The effects of spatial distribution and initial frequency of competitors on coexistence were also examined. The paper is written very well. And the 'predominance of the weakest' finding is novel.

The authors consider this as a study of biodiversity maintenance mechanisms. I feel happy with this. Meanwhile, I do think that the 'predominance of the weakest' finding is more relevant to the evolution of chemical warfare strength. Producing highly toxic chemicals may not pay off in non-transitive interactions. This could be an important selective force against the evolution of lethal warfare chemicals in microbes (the authors had one sentence to discuss this). Of course, the results can be important for understanding species coexistence, but I do have some (minor) concerns for this. First, the importance of non-transitive interactions for species coexistence in nature is not yet clear (e.g. Ecology 98: 1193-1200). Second, it is unclear if the system described in the present study (each player produces one type of chemical and resistant to the chemical it produces and exactly one chemical it does not produce) will be common in natural systems. So, I think the study is publishable as it is, but could be better if redirected to the warfare strength evolution question.

Please make sure that there is not inconsistency between the following two sentences.

'an inverse relationship exists between the starting fractional occupancy of the weakest strain and its probability to takeover'

and

'Surprisingly, we observe that the higher the starting concentration of the weak strain, the larger the parameter space in which "predominance of the weakest" occurs'.

Reviewer #3:

Remarks to the Author:

In "Survival of the weakest in non-transitive asymmetric ecologies", the authors use an engineered

non-transitive (in-vitro and in-silico) system to understand under which conditions the outcome of their experiments of competition among the engineered strains was coexistence, and why. The authors show experimentally that the strain with the weakest toxin dominates, but show computationally that the outcome of an experiment depends on the spatial arrangement and initial density ratio for the strains.

Although I'm not an experimentalist, I found the experimental work (engineering of strains, initial positioning of individuals on the agar plate) quite impressive. The degree of experimental control on the system is very exciting. I also believe that the manuscript has lots of potential to be impactful and interesting for a broad audience. That being said, there are a series of very important issues that prevent me from positively support the publication of the manuscript:

1) Existing work:

The topic this work focuses on is not new. There are lots of experimental and/or computational examples in the literature of rock-paper-scissors dynamics with engineered microbes in a spatial context (and what happens when space becomes non-important). Most of these papers were motivated by the seminal work by Kerr et al. (reference 11 in the manuscript). Given how much work on the topic there is, I would have expected at least a paragraph of the discussion (and a few sentences in the introduction) that would help the reader understand how this manuscript is different from any other existing work. That would help enormously establish the novelty and importance of the current study.

However, I found that the Kerr et al paper is barely discussed in the introduction. And there's no reference to key computational counterparts (e.g. see ref [1] below) that may or may have not have set the foundation for the model the authors present. How are the experiments and results different from the Kerr et al seminal ones? How is this computational model different from others that have been developed to represent very similar systems? What is the importance (and need) of the departures in experimental and computational design, and what are the implications of those decisions?

The answers to these questions is not only important to emphasize the novelty of the current results, but can also provide important information to understand better the system.

2) Setup and interpretation of experiments:

There are several factors that are important to understand the experimental results and the dynamics of the non-transitive cycle, but that the authors did not mention in the manuscript. For example, I would have assumed that an essential first step would be to determine the relative growth rates for the strains before and after being engineered, and the relative cost of producing each toxin (or at least effect on growth rate). The authors seem to focus on the "strength" of the toxin, but they never really quantified such strength (or I missed that part). Instead, they establish that the winner produces a "stronger toxin", but without quantifying strength that statement cannot be supported. Moreover, what the authors call toxin strength probably results from a combination of whether the toxin kills a strain and how much the growth of that strain is impacted by the synthesis of its own toxin and resistance to others. Indeed, there are plenty of examples in which the rock-paper-scissors hierarchy has been set up in terms of "binary toxins" (i.e. a strain is killed or not by the toxin, as opposed to the toxin being of a certain "strength") but the toxin taking a toll on growth leads to the hierarchical relationship.

All the above should be determined with no space first (the authors mention a "liquid culture", but used to check the hierarchy of the system after they obtained their experimental results).

Also, the (huge) differences between using or not a spatially-explicit setup should have been clearly discussed in the introduction, to help the reader understand the experimental choices made by the authors.

3) Computational model:

The level of detail in the explanation of the model is insufficient to replicate their work independently. I can, with lots of effort, get to a rough understanding of the model explained through Fig3b and methods. But I don't think I would be confident enough in my interpretation to code the model myself, which means that I would not be confident enough to critically assess whether the code that the authors offer to share is reliable, or if the model is well suited to represent the experimental setup. As an example, the way the (very important) model component of toxin-driven mortality is explained in method is really confusing. If I understand well, the toxin-driven probability for a cell to die is determined by how many "enemy" cells surround it, but that probability is capped by a number determined by the "strength" of the toxin. But they way the information is presented (referring to the latter as "maximum probability of death", instead of something like "toxin mortality", and without offering any explicit mapping from "enemy density" to mortality probability there but hidden in the schematics of Fig3b) makes it very difficult to understand that part the model in detail. Actually, the rules presented in the figure do not match the description in the text (e.g. the mortality probability is not proportional to the number of "enemies" surrounding the cell, but instead is decided with a threshold).

In any case, the dynamics are very much what would be expected from the non-transitive cycle in a spatially extended system (see reference [1] below), which is reassuring. There are, however, important computational decisions that have not been explained or justified in the text and that can crucially affect the results (so I would have expected some comments or a sort of associated sensitivity analysis):

- Model components. There is very little explanation of key choices for key rules such as mortality. Why is the minimum mortality probability 0.05? Why is the neighbor density threshold 4? Why, below that threshold, the mortality probability is half of the maximum and not any other function?
- Maximum time. The authors mention "steady state", and that they used 10,000 time steps as maximum time. But they do not mention how much that means in real time (I assume one time step represents a replication event, and replication time can be obtained in the experimental system, so the mapping would not be difficult given the right information). Without that time, we cannot really understand how the experiments and simulations compare. Did the experimental system reach stationarity? Actually, did the computational system reach stationarity (i.e. how did the authors assess that 10,000 steps were enough for all replicates to reach a steady state)? I can see in the density plots shown in the text that the simulations did, but the authors should justify their choice and explain their criterion explicitly. Also, did the authors take into account that transitory regimes to reach the steady state are going to be likely increased when the strains show similar "toxin strength"?
- Computational size and system shape. The authors used a 150x150 pixel system, but they do not mention how that compares to the real system (a pixel represents a cell, I assume, but how many cells fit in the diameter of the experimental plates?). Also, it should be clear throughout the manuscript that the computational system was always a squared lattice. Figs2 and 3 show trimmed snapshots of the system to make it look circular, for easier visual comparison with the experiments. However, and given that the system size is quite small and that the boundary conditions are not periodic (see next comment), the shape of the system is really important. Because the boundaries are absorbing (I think, see below), I would not expect important differences in the spatial pattern for square versus circular systems. But again, those non-trivial decisions need to be justified explicitly, and the consequences should have been checked systematically.

- Boundary conditions. The authors mention briefly that they reflect what is expected from experiments, and in Methods they refer to it as "zero boundary conditions", but they do not mention what exactly those conditions are or how to implement them. I am assuming they are absorbing conditions, but for the sake of reproducibility the authors should explain it explicitly.

- Parametrization. As stated above, there are lots of parameter values that the authors should have justified by mapping from the real to the in-silico system (growth rates/time step, pixel size, maximum time...). One very important parameter is the relative toxin-induced mortality, which is key because it determines how fast a strain can "propagate" in the patch of their toxin-susceptible strain. How did you translate toxicity into your parameters? Because it's a highly non-trivial task to i) quantify strength experimentally; ii) translate that to a representative model component, and iii) parametrize that component, and I would have expected all that information to be present somewhere predominantly in the manuscript.

- Initial conditions. I'm not really sure that I understand what the authors refer to when they mention "homogeneous initial distributions". To me, that would mean that there's no structure at all in the system, which would be achieved by placing at random a number of cells for each strain that follows a 1:1:1 ratio (it would also apply for other ratios, as long as the cells are placed at random). However, Figs 2a, 2d, S7a show initial arrangements for model and experiment that are not homogeneous but rather a very non-trivial, anisotropic structure in which we can define bands of cells of the same strain. I may have misinterpreted the explanation of initial conditions but, given how important these are for the final result...can the authors comment on what happens if cells are placed initially at random?

4) Manuscript in general:

I found that key terms that the authors use to informally refer to the strains are quite confusing. As an example, the term "enemy" is never defined, and I guess it means that if A produces the toxin that kills B, then A is enemy of B; it's also misleading, since a strain is not actively opposed or hostile against the other strains (just produces a toxin, regardless of what the toxin does).

Similarly, there are other never-defined terms like "attack", "competition diversity", use of the robot to "spot" the strains, "kill curve experiments", "toxin parameter", "passage", what a "liquid culture" means exactly...There are also symbols that are used in e.g. figures but not defined in the text or captions (e.g. pG, etc), and use of informal language in the text ("it's", "didn't") and figures ("1k", "10k" instead of power or scientific notation).

There are also terms that are used in a rather confusing/misleading way. For example, the introduction seems to suggest that non-transitive interactions are not hierarchical, while it's actually an example of strongly hierarchical/structured set of interactions (the authors later in the manuscript refer repeatedly to "the resulting hierarchy" in the system and the "hierarchy of toxin strengths"). Also, at some point the authors seem to refer to a "rock-paper-scissors" type of interaction for a 2-strain system, which is obviously not possible.

Finally, the Discussion section seemed very superficial, mostly reading as an extended (but not much!) summary of the results. Given all the results presented in the manuscript, what is the general interpretation and how are they going to change what we knew about the system? (which is quite a bit, see my first point above). What are the evolutionary implications of the authors' findings? How is the approach transformative? How do the lessons learned with the computational model going to change (or would have changed) the experimental setup or offer new avenues to explore?

In summary, I found the work impressive in many ways and with lots of potential, but the lack of contextualization within previous work and the lack of detail when explaining the model ultimately obscure and prevent an interesting discussion of the ecological importance and transformative potential of the main findings.

References:

[1] Luo-Luo Jiang, Tao Zhou, Matjaž Perc, and Bing-Hong Wang, "Effects of competition on pattern formation in the rock-paper-scissors game", *Phys. Rev. E* 84, 021912 (2011).

Response to Referee Critiques

General Comments to the Referees

We are grateful for their support of the quality and relevance of our study. The specific suggestions were extremely valuable in improving the impact and context for the work.

Response to Referee #1

Reviewer 1 (Remarks to the Author):

Liao et al. used engineered E. coli strains engaged in different types of colicin-mediated warfare (inhibiting protein production, digesting genomic DNA, disrupting membrane) to study cyclic Rock-Paper-Scissors interaction networks in a spatially-structured environment. As expected, the coexistence of the three strains depended on initial densities (lower density means more spatial structure and longer period of coexistence). Surprisingly, the weakest strain often dominated. This was because the weakest strain allowed its target to expand and eliminate the weakest strain's enemy, and afterwards, the weakest strain could destroy its target and dominate the entire community. The outcome of species coexistence also depended on spatial configurations. This work complements our current understanding of cyclic interactions in a spatially-structured environment. The work is interesting, and writing is clear. I have no major concerns.

We thank the reviewer for their high opinion of our work and the useful suggestion to improve the impact of our findings.

Minor: 1. Fig 4d legend: should it be strain "B" wins?

We thank the reviewer for pointing this out. Yes, it should have been strain B wins, we corrected the typo in the revised version.

2. In Discussions, "Surprisingly, we observe that xxx": this statement seems to be opposite to Fig 10?

We realized how this statement might have been confusing. In this sentence we were referring to the starting fractional concentration of the weak strain (in terms of relative percentage with respect to the other two). Therefore, we added the word "fractional" to clarify that we are not simply referring to the initial density of the array but to the relative proportion of each strain.

Review prepared by Wenying Shou

Response to Referee #2

Reviewer 2 (Remarks to the Author):

The present study shows that, in a non-transitive competitive interaction involving chemical warfare, the community is more likely to be dominated by the species (strain) with the lowest toxic strength. The effects of spatial distribution and initial frequency of competitors on co-existence were also examined. The paper is written very well. And the "predominance of the weakest" finding is novel.

The authors consider this as a study of biodiversity maintenance mechanisms. I feel happy with this. Meanwhile, I do think that the “predominance of the weakest” finding is more relevant to the evolution of chemical warfare strength. Producing highly toxic chemicals may not pay off in non-transitive interactions. This could be an important selective force against the evolution of lethal warfare chemicals in microbes (the authors had one sentence to discuss this). Of course, the results can be important for understanding species coexistence, but I do have some (minor) concerns for this. First, the importance of non-transitive interactions for species coexistence in nature is not yet clear (e.g. Ecology 98: 1193-1200). Second, it is unclear if the system described in the present study (each player produces one type of chemical and resistant to the chemical it produces and exactly one chemical it does not produce) will be common in natural systems. So, I think the study is publishable as it is, but could be better if redirected to the warfare strength evolution question.

We thank the reviewer for their strong support of the manuscript along with their perspective and suggestions below that have improved the quality of the study. First, we agree on the importance of the predominance of the weakest in the context of the evolution of chemical warfare. To address this, we expanded on this topic in the discussion section by adding the following paragraph: “Counterintuitively, under the same initial conditions of uniform distribution, our models predict that the producer of the most lethal toxin never wins. As opposed to pairwise competition, where the producer of the strongest toxin has a competitive advantage, the producer of the strongest toxin in non-transitive communities are at an evolutionary disadvantage. This could be an important selective force against continuous evolution of ever-more lethal warfare chemicals in microorganisms, resulting in increased diversity of chemicals that are constrained to a specific toxin strength parameter space. The role that toxin mediated competition plays in community stability may also explain the observed relative abundance of membrane targeting,

DNA, or ribosome targeting bacterial toxins among bacterial communities. ”

Furthermore, we thank the reviewer for bringing to our attention the relevant study about the importance of intransitive competition in nature (Ecology 98: 1193-1200). We cited it into the discussion section in the following paragraph: While previous studies focused on the coexistence of non-transitive communities over a relatively short timeframe, they do not allow the system to reach steady state, in which case it is likely that only one species would remain. Here, we demonstrate that intransitivity fails to promote biodiversity over a long time horizon when the relative competitive advantages are imbalanced. Therefore, we believe that an asymmetric non-transitive ecology is a useful base model to study complex interactions among competing bacterial species.

Please make sure that there is not inconsistency between the following two sentences. “an inverse relationship exists between the starting fractional occupancy of the weakest strain and its probability to takeover” and “Surprisingly, we observe that the higher the starting concentration of the weak strain, the larger the parameter space in which “predominance of the weakest” occurs”.

We thank the reviewer for pointing out this inconsistency. The same response we used to answer Reviewer 1 question is valid here. We realized how this statement might have been confusing. In this sentence (Surprisingly, we observe that) we were referring to the starting fractional concentration of the weak strain (in terms of relative percentage with respect to the other two). Therefore, we added the word “fractional” to clarify that we are not simply referring to the initial density of the seeding.

Response to Referee #3

Reviewer 3 (Remarks to the Author):

In “Survival of the weakest in non-transitive asymmetric ecologies”, the authors use an engineered non-transitive (in-vitro and in-silico) system to understand under which conditions the outcome of their experiments of competition among the engineered strains was coexistence, and why. The authors show experimentally that the strain with the weakest toxin dominates, but show computationally that the outcome of an experiment depends on the spatial arrangement and initial density ratio for the strains.

Although I’m not an experimentalist, I found the experimental work (engineering of strains, initial positioning of individuals on the agar plate) quite impressive. The degree of experimental control on the system is very exciting. I also believe that the manuscript has lots of potential to be impactful and interesting for a broad audience. That being said, there are a series of very important issues that prevent me from positively support the publication of the manuscript:

1) Existing work:

The topic this work focuses on is not new. There are lots of experimental and/or computational examples in the literature of rock-paper-scissors dynamics with engineered microbes in a spatial context (and what happens when space becomes non-important). Most of these papers were motivated by the seminal work by Kerr et al. (reference 11 in the manuscript). Given how much work on the topic there is, I would have expected at least a paragraph of the discussion (and a few sentences in the introduction) that would help the reader understand how this manuscript is different from any other existing work. That would help enormously establish the novelty and importance of the current study.

However, I found that the Kerr et al paper is barely discussed in the introduction. And there's no reference to key computational counterparts (e.g. see ref [1] below) that may or may have not have set the foundation for the model the authors present. How are the experiments and results different from the Kerr et al seminal ones? How is this computational model different from others that have been developed to represent very similar systems? What is the importance (and need) of the departures in experimental and computational design, and what are the implications of those decisions?

The answers to these questions is not only important to emphasize the novelty of the current results, but can also provide important information to understand better the system.

We want to thank Reviewer 3 for his positive opinion of our study and the in-depth comments which helped us improve the manuscript greatly. We agree that previous work could be expanded further, therefore we added several paragraphs to analyze previous work in comparison to our study. In the introduction we added the following paragraph: “While the application of theory in ecology is still limited, the principles of a non-transitive ecology were first demonstrated experimentally using three isolated bacterial strains consisting of a toxin producing, toxin sensitive, and toxin resistant strain. In this ecology, the toxin producer could kill the toxin sensitive strain, the toxin sensitive strain could outgrow the toxin resistant strain, likewise the toxin resistant strain could outgrow the toxin producing strain. This simple, non-transitive triplet demonstrated that when interactions among the strains remained local, cyclic competition could maintain species diversity. However, this study only focused on a short observation interval of seven days and did not provide a characterization of the relative competitive advantages of each strain. For example, it seems intuitive that a toxin producing strain is more likely

to outcompete a toxin-sensitive strain at a faster rate than a toxin sensitive strain could outgrow a toxin resistant strain. Due to this potential asymmetry within this system, it seems unlikely that coexistence would be maintained over a longer experimental duration. To explore these hypotheses, we investigate how asymmetry affects a three strain cyclic ecology, expanding upon the previous experimental work using rationally engineered and well characterized strains of *E. coli*. Furthermore, we develop a computational model to thoroughly investigate the relationship between asymmetric competition in relation to multiple parameters, such as species composition, relative competitive strengths, and spatial patterns.”

2) *Setup and interpretation of experiments:*

There are several factors that are important to understand the experimental results and the dynamics of the non-transitive cycle, but that the authors did not mention in the manuscript. For example, I would have assumed that an essential first step would be to determine the relative growth rates for the strains before and after being engineered, and the relative cost of producing each toxin (or at least effect on growth rate).

We thank the reviewer for pointing out the need to include the characterization of the growth rate among engineered. We added this information in the supplementary material (Extended Data Figure 4 and 6).

The authors seem to focus on the "strength" of the toxin, but they never really quantified such strength (or I missed that part). Instead, they establish that the winner produces a "stronger toxin", but without quantifying strength that statement cannot be supported. Moreover, what the authors call toxin strength probably results from a combination of whether the toxin kills a strain and how much the growth of that strain is impacted by the synthesis of its own

toxin and resistance to others. Indeed, there are plenty of examples in which the rock-paper-scissors hierarchy has been set up in terms of "binary toxins" (i.e. a strain is killed or not by the toxin, as opposed to the toxin being of a certain "strength") but the toxin taking a toll on growth leads to the hierarchical relationship. All the above should be determined with no space first (the authors mention a "liquid culture", but used to check the hierarchy of the system after they obtained their experimental results).

We thank the reviewer for this comment. We agree on the importance of defining "toxin strength" and the "relative toxin strength". In this work we focused on characterizing the relative toxin efficacy (relative toxin strength) among each strain pair within RPS community. The method we used to find the relative toxin efficacies in liquid culture is described in the method section named "Kill Curve". In addition, we also verified the same relationship on agar plates by testing pairs of toxin producer/toxin sensitive strains and counting the number of days needed for the producer to completely takeover. The results from the first method are shown in Extended Data Figure 4 and 6, while the results from the second method are included in Figure 2 in the main text.

Additionally, while there may be complicated coupling of effects between the toxins synthesis, growth rate, and sensitivity towards other toxins, our goal was to develop a framework for understanding competition dynamics that are relative to the species in the community. Because we did not see any major differences in growth rate among the three strains, we attributed any reduction in competitive fitness to the effects of the toxin produced by the dominant strain in each pair. Therefore, we feel that the liquid kill curve experiments, as well as the paired agar plate competition experiments were sufficient in establishing the relative hierarchy.

Also, the (huge) differences between using or not a spatially-explicit setup should have been

clearly discussed in the introduction, to help the reader understand the experimental choices made by the authors.

We thank the reviewer for pointing out the need to clearly state the reason behind our experimental set-up choices. Therefore, we added the following paragraph to the Results section: “The choice to carry out all our experiments on solid media was based on the previously reported observation that local interactions and dispersal promote diversity within a community”.

3) Computational model:

The level of detail in the explanation of the model is insufficient to replicate their work independently. I can, with lots of effort, get to a rough understanding of the model explained through Fig3b and methods. But I don't think I would be confident enough in my interpretation to code the model myself, which means that I would not be confident enough to critically assess whether the code that the authors offer to share is reliable, or if the model is well suited to represent the experimental setup. As an example, the way the (very important) model component of toxin-driven mortality is explained in method is really confusing. If I understand well, the toxin-driven probability for a cell to die is determined by how many "enemy" cells surround it, but that probability is capped by a number determined by the "strength" of the toxin. But they way the information is presented (referring to the latter as "maximum probability of death", instead of something like "toxin mortality", and without offering any explicit mapping from "enemy density" to mortality probability there but hidden in the schematics of Fig3b) makes it very difficult to understand that part the model in detail. Actually, the rules presented in the figure do not match the description in the text (e.g. the mortality probability is not proportional to the number of "enemies" surrounding the cell, but instead is decided with a threshold).

We thank the reviewer for his very key and insightful comments on the details about the model description. We made several efforts to address these concerns. First, we expanded our description in the method section to address reproducibility of the model. Furthermore, we modified the description in the method section in order to be consistent with Figure 3b. We confirm that the reviewer understanding of the model is correct as expressed in this statement: the toxin-driven probability for a cell to die is determined by how many "enemy" cells surround it, but that probability is capped by a number determined by the "strength" of the toxin.

In any case, the dynamics are very much what would be expected from the non-transitive cycle in a spatially extended system (see reference [1] below), which is reassuring. There are, however, important computational decisions that have not been explained or justified in the text and that can crucially affect the results (so I would have expected some comments or a sort of associated sensitivity analysis):

- Model components. There is very little explanation of key choices for key rules such as mortality. Why is the minimum mortality probability 0.05? Why is the neighbor density threshold 4? Why, below that threshold, the mortality probability is half of the maximum and not any other function?

We thank the reviewer for bringing these questions to our attention. We include a minimum mortality parameter in order to account for several variables that lead to cell death or removal that is independent of death due to toxin competition. These include the probability of cell death due to the production of its own toxin, death upon expression of the Colicin E1 lysis protein for toxin release, as well as experimental factors such as loss of cell during the replica plating process. This process consists of using a velvet pad to transfer cells between plates. We

experimented with a range of parameter values and used the one that was best able to reproduce the experimental outcomes.

In terms of the mortality probability function that is associated with death from toxin produced by competitors, we decided to use a threshold dependency to reflect that cell death occurs when a certain toxin threshold (LD50) is reached, rather than being a linear function related to toxin concentration. This is represented in our mathematical model by setting the mortality probability at half of maximum when the toxin concentration is low (neighbor density is less than 4). Alternatively, if the enemies occupy greater than half of the surrounding pixels, then we set the mortality probability at a maximum. In both cases the toxin can be lethal, however the probability of death is not linear, but rather flips according to a threshold. We are aware that previous models in the literature used a linear relationship where the probability of death was linearly correlated to the number of enemies surrounding the cells, however, we found that this threshold function better represents the experimental observations.

- Maximum time. The authors mention "steady state", and that they used 10,000 time steps as maximum time. But they do not mention how much that means in real time (I assume one time step represents a replication event, and replication time can be obtained in the experimental system, so the mapping would not be difficult given the right information). Without that time, we cannot really understand how the experiments and simulations compare. Did the experimental system reach stationarity? Actually, did the computational system reach stationarity (i.e. how did the authors assess that 10,000 steps were enough for all replicates to reach a steady state)? I can see in the density plots shown in the text that the simulations did, but the authors should justify their choice and explain their criterion explicitly.

We thank the reviewer for helping us clarify this. For the simulations shown in Figure 4,

each simulation was run for $t=10000$ and each parameter set was simulated 100 times. Each time point corresponds to a reproduction event (around 25 minutes). Therefore, the time chosen to investigate steady state dynamics corresponds to about 170 days, which we established to be a long enough interval both computationally and experimentally.

Also, did the authors take into account that transitory regimes to reach the steady state are going to be likely increased when the strains show similar "toxin strength"?

Yes, we considered a time interval that was much longer than the maximum time recorded to reach steady state from our single time series observations. As the reviewer pointed out, the simulations with strains of similar toxin strengths take longer to reach steady state. Therefore, this is the case we took into consideration when deciding which time interval to use to investigate the steady state variations in the parameter space chosen.

- Computational size and system shape. The authors used a 150x150 pixel system, but they do not mention how that compares to the real system (a pixel represents a cell, I assume, but how many cells fit in the diameter of the experimental plates?).

In terms of spatial parameters, the distance between two consecutive pixels represents roughly 1 mm on the agar plate. Therefore, the entire grid represents a square with a side of about 15 cm. The relationship between grid densities on the agar plates compared to the model are illustrated in Extended Data Fig. 7. We consider each pixel as a 2D colony of a given strain, comparable to the colonies that we spot using the liquid handler.

Also, it should be clear throughout the manuscript that the computational system was always

a squared lattice. Figs2 and 3 show trimmed snapshots of the system to make it look circular, for easier visual comparison with the experiments. However, and given that the system size is quite small and that the boundary conditions are not periodic (see next comment), the shape of the system is really important. Because the boundaries are absorbing (I think, see below), I would not expect important differences in the spatial pattern for square versus circular systems. But again, those non-trivial decisions need to be justified explicitly, and the consequences should have been checked systematically.

As the reviewer observed, all simulations were executed on a square lattice. Fig 2 only contains experimental figures, while Fig 3 contains both experimental results and model simulations. In particular, the lattice simulations were trimmed (post-simulation) to make the comparison with experimental data easier. Therefore, we thank the reviewer suggestion and added the following sentence to the figure caption: All lattice simulations plots were simulated in a square lattice grid and subsequently trimmed into circular shapes to ease the comparison with experimental results.

The full simulations are included as videos. We agree with the reviewer that since the boundaries are absorbing, we would not expect important differences in the spatial pattern for square versus circular systems.

- Boundary conditions. The authors mention briefly that they reflect what is expected from experiments, and in Methods they refer to it as "zero boundary conditions", but they do not mention what exactly those conditions are or how to implement them. I am assuming they are absorbing conditions, but for the sake of reproducibility the authors should explain it explicitly.

The boundary conditions were chosen to reflect the physical boundary of a confined space

such as the petri dish we use for the experiments in this paper. Therefore, the four edges of the square are kept at a value of zero (no cells can grow/expand in that direction) and the grid is updated only for the internal pixel (pixel 2 to N-1 on all sides). We added this information in the Method section. This is one of the main differences compared to previous models from the literature which used periodic boundary conditions.

- Parametrization. As stated above, there are lots of parameter values that the authors should have justified by mapping from the real to the in-silico system (growth rates/time step, pixel size, maximum time...). One very important parameter is the relative toxin-induced mortality, which is key because it determines how fast a strain can "propagate" in the patch of their toxin-susceptible strain. How did you translate toxicity into your parameters? Because it's a highly non-trivial task to i) quantify strength experimentally; ii) translate that to a representative model component, and iii) parametrize that component, and I would have expected all that information to be present somewhere predominantly in the manuscript.

We thank the reviewer for pointing this out as determining toxin parameters was among our major focuses. In order to quantify the relative strength between strains we used the data obtained from the kill curve experiment which is described in the Methods section. In particular, we took the difference of 5 time points (40 minutes) that corresponded to the maximum change in total alive cell count divided by the initial CFU count of the initial time point in order to calculate the percent cell death. The relative toxin strengths were then found by taking the ratios between the percent cell deaths for each strain pair, with the minimum toxin strength being normalized to one. This information is reported in the Kill Curve section of the Methods section.

- Initial conditions. I'm not really sure that I understand what the authors refer to when they

mention "homogeneous initial distributions". To me, that would mean that there's no structure at all in the system, which would be achieved by placing at random a number of cells for each strain that follows a 1:1:1 ratio (it would also apply for other ratios, as long as the cells are placed at random). However, Figs 2a, 2d, S7a show initial arrangements for model and experiment that are not homogeneous but rather a very non-trivial, anisotropic structure in which we can define bands of cells of the same strain. I may have misinterpreted the explanation of initial conditions but, given how important these are for the final result...can the authors comment on what happens if cells are placed initially at random?

We thank the reviewer for pointing this out, we completely agree how our definition of initial condition pattern might have been confusing. By the term "uniform" or "homogeneous" distribution we refer to the fact that the three strains are assembled in a grid pattern that is uniformly repeated across the plate. Therefore, we clarified this statement by changing this sentence: "Therefore, we designed an experimental protocol which could guarantee uniform grid-like initial spatial distribution of the three strains."

On the other hand, the experiment that shows what happened when the cells are randomly plated on the agar plate (instead of being placed in a grid pattern) is shown in Extended Data Figure 2. The experiment shows that the outcome of the RPS competition is the same as the one for the grid pattern experiment (Strain B wins).

4) Manuscript in general:

I found that key terms that the authors use to informally refer to the strains are quite confusing. As an example, the term "enemy" is never defined, and I guess it means that if A produces the toxin that kills B, then A is enemy of B; it's also misleading, since a strain is not actively opposed or hostile against the other strains (just produces a toxin, regardless of what the toxin

does).

Similarly, there are other never-defined terms like "attack", "competition diversity", use of the robot to "spot" the strains, "kill curve experiments", "toxin parameter", "passage", what a "liquid culture" means exactly...There are also symbols that are used in e.g. figures but not defined in the text or captions (e.g. pG, etc), and use of informal language in the text ("it's", "didn't") and figures ("1k", "10k" instead of power or scientific notation).

We thank the reviewer for the pointing out these inconsistencies. In particular, we changed the verb "spot" to "array" to indicate the act of positioning cells onto agar plates using the liquid handler. Furthermore, we got rid of all use of informal language and substituted the "1k" format into scientific notation. In addition, we added a description of the acronyms (pG,pB,pR) in the Modeling part of the Method section.

There are also terms that are used in a rather confusing/misleading way. For example, the introduction seems to suggest that non-transitive interactions are not hierarchical, while it's actually an example of strongly hierarchical/structured set of interactions (the authors later in the manuscript refer repeatedly to "the resulting hierarchy" in the system and the "hierarchy of toxin strengths"). Also, at some point the authors seem to refer to a "rock-paper-scissors" type of interaction for a 2-strain system, which is obviously not possible.

We thank the reviewer for pointing out this misuse of the term "hierarchy". We refer to cyclic interactions such as rock-paper-scissor dynamics as non-hierarchical because there is no species at the top of the predation chain. We understand that the term hierarchy used to describe relative toxin strength was misleading, therefore we changed it to the term "ranking" or "relative toxin strengths" as in the following sentence: "The resulting ranking of relative toxin

strengths was also demonstrated in kill curve experiments conducted in liquid culture”.

Finally, the Discussion section seemed very superficial, mostly reading as an extended (but not much!) summary of the results. Given all the results presented in the manuscript, what is the general interpretation and how are they going to change what we knew about the system? (which is quite a bit, see my first point above). What are the evolutionary implications of the authors' findings? How is the approach transformative? How do the lessons learned with the computational model going to change (or would have changed) the experimental setup or offer new avenues to explore?

We thank the reviewer for comments helping us improve the quality of the discussion. We expanded the discussion in the manuscript to address these points as follows:

“Bacterial communities occupy a myriad of diverse niche environments, playing important roles in processes ranging from nutrient recycling to the regulation of human health. While the ability to engineer robust bacterial communities could lead to major advancements in fields such as recycling, sustainability, and healthcare, the mechanisms underlying species diversity and stability are still not yet well understood. Although cooperative interactions between the species comprising such communities contribute to ecological stability, it is generally accepted that competition is the guiding force. However, due to the complexity of natural ecologies, these types of competitive interactions are difficult to isolate or quantify in nature. Our study demonstrates the feasibility of using an engineered synthetic ecology to simplify complex community relationships in order to study underlying mechanisms that may lead to community stability and the maintenance of diversity.

Due to the immense species diversity and wide range of competitive strategies organisms

employ in nature, we hypothesized that natural competitive dynamics are likely to be unbalanced. Unlike a perfectly balanced game of rock-paper-scissors in which each of the three species could kill each other at an equal rate, we focus on characterizing an asymmetric system in which the relative competitive advantages among each predator-prey pair varies. While previous studies focused on the coexistence of non-transitive communities over a relatively short timeframe, they do not allow the system to reach steady state, in which case it is likely that only one species would remain. Here, we demonstrate that intransitivity fails to promote biodiversity over a long time horizon when the relative competitive advantages are imbalanced. Therefore, we believe that an asymmetric non-transitive ecology is a useful base model to study complex interactions among competing bacterial species. Using our three-strain ecology we experimentally demonstrate that a uniformly distributed asymmetric game of rock-paper-scissor is most likely to be won by the weakest species. Interestingly, we show that an asymmetric ecology can develop steady state coexistence, and that the relative toxin strengths among the three species dictate the extent of the coexistence space. Counterintuitively, under the same initial conditions of uniform distribution, our models predict that the producer of the most lethal toxin never wins. As opposed to pairwise competition, where the producer of the strongest toxin has a competitive advantage, the producer of the strongest toxin in non-transitive communities are at an evolutionary disadvantage. This could be an important selective force against continuous evolution of ever-more lethal warfare chemicals in microorganisms, resulting in increased diversity of chemicals that are constrained to a specific toxin strength parameter space. The role that toxin mediated competition plays in community stability may also explain the observed relative abundance of membrane targeting, DNA, or ribosome targeting bacterial toxins among bacterial communities. Finally, we also observe that the steady-state outcomes of the system can be altered by changing the initial strain distribution patterns. For example, we find that separate blocks in a triangular conformation can greatly expand the parameter space for coexistence,

supporting the idea that spatially separated niches are more likely to sustain biodiversity. On the other hand, strains initially distributed in concentric circles can enable the strongest toxin producer to win. These results demonstrate that many factors need to be considered if the goal is to design stable synthetic ecologies in an environment where interactions are local. Overall, this study provides a mathematical model and engineering framework to study competitive interactions, gain mechanistic insight, and ultimately, predictive power that can be used as a guide to design stable communities.”

In summary, I found the work impressive in many ways and with lots of potential, but the lack of contextualization within previous work and the lack of detail when explaining the model ultimately obscure and prevent an interesting discussion of the ecological importance and transformative potential of the main findings.

We thank the reviewer for their insightful comments that helped us improve the quality and impact of this study.

Reviewers' Comments:

Reviewer #3:

Remarks to the Author:

I found the revised version of the manuscript quite improved, and that it addresses almost all the issues raised by the referees. I'm especially pleased with the Discussion section now really connects the results with their ecological implications, and the explanation of the model. Regarding the latter, however, there are still a few things that need clarification.

One concern that still remains, though, is the choice of boundary conditions. The authors mention that sites at the border are kept to 0. This could be interpreted as "cells can replicate to any side even at the borders of the system, but if the offspring are/would be placed at or outside the border, they are not considered", which would correspond to absorbing boundary conditions. In a petry dish, however, what happens is that offspring from cells at the border cannot be placed at particular places, only inside the system, i.e. the cell cluster cannot expand in one direction but instead expands faster in the opposite one as the result of the offspring being always placed inside the system (i.e. never lost). The latter corresponds to reflecting boundary conditions, which have a huge effect on the dynamics if the system is not large enough.

The authors should clarify which of the two were really chosen and, if absorbing, how a different choice (reflecting, or even periodic to minimize border effects) would alter their results. Either way, that would not be a problem for sufficiently large systems, but for the system sizes used here it is important to make the clarification.

Other than that, I just have a couple of recommendations to make the manuscript more user-friendly and approachable:

- One is the definition of the initial conditions. I still find the use of "uniform" confusing, and maybe the authors could include an additional sentence explaining that it refers to the fact that the overall densities are the same for the three strains, but the system is actually initialized in a spatially heterogeneous way. I thought that the "grid" clarification was not enough.
- Another is the inclusion of a table that compiles all parameters used in the model, a brief definition, their value, and one sentence pointing to what experimental aspect informs that decision. It of course would be expected to be an Extended Table, but I think it's good to have that one place where anyone that ones to replicate the model results can go, as well as to connect model and experiments.

As I said, I think the boundary conditions issue is a very necessary clarification to make, but if after the authors respond to my comments the editor thinks that the manuscript is ready for publication, I do not need to see the new revised version.

Response to Referee Critiques (Round 2)

General Comments to the Referees

We thank Reviewer 3 for his very useful comments and the attention to details that allowed us to further improve our manuscript.

Response to Reviewer #3

Reviewer #3 (Remarks to the Author):

I found the revised version of the manuscript quite improved, and that it addresses almost all the issues raised by the referees. I'm especially pleased with the Discussion section now really connects the results with their ecological implications, and the explanation of the model. Regarding the latter, however, there are still a few things that need clarification.

One concern that still remains, though, is the choice of boundary conditions. The authors mention that sites at the border are kept to 0. This could be interpreted as "cells can replicate to any side even at the borders of the system, but if the offspring are/would be placed at or outside the border, they are not considered", which would correspond to absorbing boundary conditions. In a petry dish, however, what happens is that offspring from cells at the border cannot be placed at particular places, only inside the system, i.e. the cell cluster cannot expand in one direction but instead expands faster in the opposite one as the result of the offspring being always placed inside the system (i.e. never lost). The latter corresponds to reflecting boundary conditions, which have a huge effect on the dynamics if the system is not large enough. The authors should clarify which of the two were really chosen and, if absorbing, how a different choice (reflecting, or even periodic to minimize border effects) would alter their results. Either

way, that would not be a problem for sufficiently large systems, but for the system sizes used here it is important to make the clarification.

We thank the reviewer for his comments. We confirm that the boundary conditions chosen are absorbing and can indeed be interpreted as "cells can replicate to any side even at the borders of the system, but if the offspring are/would be placed at or outside the border, they are not considered". Accordingly, we edited the model section of the manuscript to clearly state that we are using absorbing boundary conditions as follows : "The lattice is a 150 x 150 regular square lattice with zero boundary conditions. This means that the edges of the lattice are set to zero (**absorbing boundary conditions**), simulating the physical boundary of the petri dish that prevents cells from expanding beyond it as well as the effect of disregarding cells beyond the boundary of the replica plating." As the reviewer pointed out, if the system considered is large enough the choice of boundary conditions doesn't significantly influence the overall dynamics. Our model grid simulates a physical space of side 10 cm (the petri dish) while each cell has dimension of the micrometer order. Therefore, we considered absorbing boundary conditions to be better suited to replicate the experimentally observed dynamics in a computational grid of size 150 pixels. In addition, the choice of absorbing boundary conditions was justified by the effect of replica plating which only transfers cells within the boundary of the velvet pad which has a diameter slightly smaller than the petri dish surface.

Other than that, I just have a couple of recommendations to make the manuscript more user-friendly and approachable:

- One is the definition of the initial conditions. I still find the use of "uniform" confusing, and maybe the authors could include an additional sentence explaining that it refers to the fact that the overall densities are the same for the three strains, but the system is actually initialized in a

spatially heterogeneous way. I thought that the "grid" clarification was not enough.

We thank the reviewer for pointing out the need for clarification on the concept of the grid format. We therefore edited our previous sentence and rephrased it as follows: "Therefore, we designed an experimental protocol which could guarantee the sequential placement of the three strains in individual colonies which are equidistant from each other is a grid format."

- Another is the inclusion of a table that compiles all parameters used in the model, a brief definition, their value, and one sentence pointing to what experimental aspect informs that decision. It of course would be expected to be an Extended Table, but I think it's good to have that one place where anyone that ones to replicate the model results can go, as well as to connect model and experiments.

We thank the reviewer for this useful suggestion. We added a supplementary table that includes all the parameters used in the computational model.

As I said, I think the boundary conditions issue is a very necessary clarification to make, but if after the authors respond to my comments the editor thinks that the manuscript is ready for publication, I do not need to see the new revised version.